# SoLar: Sinkhorn Label Refinery for Imbalanced Partial-Label Learning

**Haobo Wang**[1]    **Mingxuan Xia**[2]    **Yixuan Li**[3]    **Yuren Mao**[2]
**Lei Feng**[45]    **Gang Chen**[1]    **Junbo Zhao**[1*]

[1]Key Lab of Intelligent Computing based Big Data of Zhejiang Province, Zhejiang University
[2]School of Software Technology, Zhejiang University
[3]Department of Computer Sciences, University of Wisconsin-Madison
[4]College of Computer Science, Chongqing University
[5]Center for Advanced Intelligence Project, RIKEN
`{wanghaobo,xiamingxuan,yuren.mao,cg,j.zhao}@zju.edu.cn`
`sharonli@cs.wisc.edu, lfeng@cqu.edu.cn`

## Abstract

*Partial-label learning* (PLL) is a peculiar weakly-supervised learning task where the training samples are generally associated with a set of candidate labels instead of single ground truth. While a variety of label disambiguation methods have been proposed in this domain, they normally assume a class-balanced scenario that may not hold in many real-world applications. Empirically, we observe degenerated performance of the prior methods when facing the combinatorial challenge from the long-tailed distribution and partial-labeling. In this work, we first identify the major reasons that the prior work failed. We subsequently propose **SoLar**, a novel Optimal Transport-based framework that allows to refine the disambiguated labels towards matching the marginal class prior distribution. SoLar additionally incorporates a new and systematic mechanism for estimating the long-tailed class prior distribution under the PLL setup. Through extensive experiments, SoLar exhibits substantially superior results on standardized benchmarks compared to the previous state-of-the-art PLL methods. Code and data are available at: `https://github.com/hbzju/SoLar`.

## 1 Introduction

The remarkable success of deep learning typically requires collecting massive labeled data, which is notoriously labor-intensive. Of particular concern, data labeling can suffer from inherent and pervasive label ambiguity. Take Figure 1 (a) as an example, a Mule can be visually similar to both Donkeys and Horses, which hinders non-expert annotators from recognizing the true breed. Similar cases also arise in many real-world applications, such as automatic image annotation [1], bird song recognition [2], and facial age estimation [3]. To deal with ambiguous supervision, *partial-label learning* (PLL) [4, 5], which allows each training example to be annotated by a set of candidate labels, has attracted significant attention from the community. A plethora of methods have been developed to tackle this problem, including pseudo-labeling methods [6, 7], graph-based methods [8, 9, 10], etc.

Despite the promise, existing PLL methods have been commonly driven by the assumption that training data consists of class-balanced distribution, which may not hold in practice. In many real-world scenarios, training data exhibits a *long-tailed label distribution* [11]. That is, many labels occur infrequently in the training data. Concerningly, the imbalanced data can exacerbate the fundamental

---

[*]Corresponding author.

36th Conference on Neural Information Processing Systems (NeurIPS 2022).

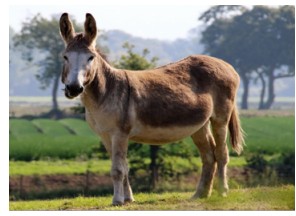

$S$ = {Donkey, Mule, Horse}

(a) Example of label ambiguity

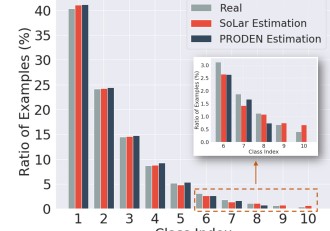

(b) Class prior estimation

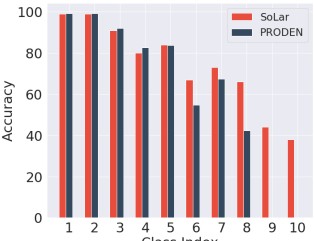

(c) Class-wise accuracy

Figure 1: (a) An input image with three candidate labels, where the ground-truth is `Mule`. (b) The real/estimated class distribution (by counting real/predicted labels) on a long-tailed version of CIFAR10 dataset with partial labels (more details in Section 4). The class prior estimated by SoLar is very close to the real one, while PRODEN fails to recognize some tail labels. (c) Class-wise performance comparison of PRODEN and SoLar on the same dataset, where SoLar performs much better on data-scarce classes.

challenge of label disambiguation, *i.e.*, identifying the ground-truth label from candidates. Indeed, we find that current best-performing PLL methods display degenerated performance in the long-tailed setting. This happens because the predictions of pseudo-labeling—a core component that PLL methods rely on—can be largely biased towards the head and majority classes. We exemplify the phenomenon in Figure 1 (b), where a strong method PRODEN [6] rarely assigns probability mass to the tail classes (indexed by 9 and 10). As a result, these tail classes suffer from almost zero accuracy, as shown in Figure 1 (c). To date, few efforts have been made to resolve this.

Motivated by this, we propose a novel framework for long-tailed partial-label learning, called **S**inkh**o**rn **La**bel **r**efinery (dubbed **SoLar**). Our framework emphasizes the long-tailed nature of training data, while performing label disambiguation for partial-label learning. Our key idea is to enforce constraints on pseudo-labels—the distribution of which should *match the class prior distribution*. The constraints can incentivize the label disambiguation process to select tail labels from candidate sets, instead of always selecting head classes. We formalize the idea as an optimal transport problem [12] that searches for a proper label assignment subjected to constraints: (1) the probability mass is distributed within candidate sets; (2) the distribution of (predicted) pseudo-labels matches the prior class distribution. The first constraint is natural in the PLL setup, and the latter one uniquely empowers SoLar to disambiguate tail labels from candidates. We show that our constrained optimization objective can be tractably solved using the Sinkhorn-Knopp algorithm [13], which incurs only a few computational overheads. Theoretically, we prove that our overall objective is statistically consistent, which ensures the optimality of the learned classifier at a population level.

As an integral part of our framework, SoLar tackles the challenge of estimating the class prior distribution, which is necessitated in our constrained optimization. Unlike conventional long-tailed learning (LTL) scenarios [14, 15], the PLL setup is presented with label ambiguity in the proximity of candidate sets. This makes it *intractable* to estimate the marginal class prior distribution, especially by counting training samples grouped by class labels. We tackle this non-trivial problem by performing an iterative class prior estimation, which provides a strong proxy for real prior (Section 3.2). Empirically in Figure 1 (b), we show that the estimated class prior favorably matches the ground-truth prior on a long-tailed version of the CIFAR10 dataset with partial labels.

We comprehensively evaluate SoLar on various benchmark datasets, where SoLar establishes state-of-the-art performance. Compared to the best baseline, SoLar improves the accuracy on tail classes by **15.12**% on the long-tailed version of the CIFAR10 dataset. While our work primarily focuses on label disambiguation from the PLL perspective, we demonstrate that SoLar can be compatible with LTL techniques as well. For example, we equip SoLar with logit adjustment [14], a state-of-the-art LTL method and further improve SoLar's performance by **6.72**% on long-tailed CIFAR10. We hope our work will inspire future works to tackle this important problem.

## 2 Problem Setup

Let $\mathcal{X} = \mathbb{R}^d$ denote the input space and $\mathcal{Y} = \{1, \ldots, L\}$ denote the label space. We receive a training dataset $\mathcal{D} = \{(\boldsymbol{x}_i, S_i)\}_{i=1}^n$ with $n$ examples. Each tuple in $\mathcal{D}$ consists of an image $\boldsymbol{x}_i \in \mathcal{X}$ and

**Algorithm 1:** Pseudo-code of SoLar.

1  **Input:** Training dataset $\mathcal{D}$, classifier $f$, uniform marginal $\boldsymbol{r}, \boldsymbol{c}$, hyperparameters $\rho, \tau, \mu, \lambda$.
2  **for** $epoch = 1, 2, \ldots,$ **do**
3     **for** $step = 1, 2, \ldots,$ **do**
4        Get classifier prediction $\boldsymbol{P}$ on a mini-batch of data $B$ of size $n_B$
5        Calculate $\boldsymbol{M}$ such that $m_{ij} = p_{ij}^{\lambda} \mathbb{I}(j \in S_i)$
6        **for** $t = 1, \ldots, T$ **do**
7           $\boldsymbol{\alpha} \leftarrow \boldsymbol{c}./(\boldsymbol{M}\boldsymbol{\beta}), \quad \boldsymbol{\beta} \leftarrow \boldsymbol{r}./(\boldsymbol{M}^{\top}\boldsymbol{\alpha})$        `// Sinkhorn-Knopp iteration`
8        **end**
9        $\boldsymbol{Q} = n_B \cdot \text{diag}(\boldsymbol{\alpha})\boldsymbol{M}\text{diag}(\boldsymbol{\beta})$        `// Compute pseudo-labels`
10       **for** $j = 1, \ldots, L$ **do**
11          Generate a subset $B_j = \{(\boldsymbol{x}_i, \boldsymbol{q}_i)|j = \arg\max_{j' \in S_i} q_{ij'}\}$ from $B$
12          Select reliable samples by small loss $l_i$ and high-confidence $e_i > \tau$
            `// Class-wise reliable sample selection`
13       **end**
14       Train the classifier $f$ by minimizing the cross-entropy loss
15    **end**
16    Compute $\boldsymbol{z}$ by $z_j = \frac{1}{n}\sum_{i=1}^{n} \mathbb{I}(j = \arg\max_{j' \in S_i} f_{j'}(\boldsymbol{x}_i))$   `// Empirical class prior estimation`
17    $\boldsymbol{r} \leftarrow \mu\boldsymbol{r} + (1-\mu)\boldsymbol{z}$           `// Moving-average distribution updating`
18 **end**

the candidate label set $S_i \subset \mathcal{Y}$. Following previous works [6, 7], we assume the true label $y_i \in \mathcal{Y}$ of $\boldsymbol{x}_i$ is concealed in $S_i$, *i.e.*, $y_i \in S_i$. A fundamental challenge in partial-label learning is label disambiguation, *i.e.*, identifying the ground-truth label $y_i$ from the candidate label set $S_i$.

Our goal is to train a classifier $f : \mathcal{X} \mapsto [0,1]^L$, parameterized by $\theta$, that can perform predictions on unseen testing data. Here $f$ is the softmax output of a neural network, and $f_j(\cdot)$ denotes the $j$-th entry. Besides, we denote the prediction matrix by $\boldsymbol{P} = [\boldsymbol{p}_1, \ldots, \boldsymbol{p}_n]^{\top} = [p_{ij}]_{n \times L}$, where $p_{ij} = f_j(\boldsymbol{x}_i)$. To perform label disambiguation, we maintain a pseudo-label matrix $\boldsymbol{Q} = [\boldsymbol{q}_1, \ldots, \boldsymbol{q}_n]^{\top} \in [0,1]^{n \times L}$ and train the classifier with the cross-entropy loss $l_{\text{ce}}(f; \boldsymbol{x}_i, \boldsymbol{q}_i) = \sum_{j=1}^{L} -q_{ij} \log(f_j(\boldsymbol{x}_i))$, where $q_{ij}$ is the $j$-th entry of $\boldsymbol{q}_i$.

## 3 Proposed Method

In this section, we describe our novel **S**inkh**o**rn **La**bel **r**efinery (SoLar) framework for partial-label learning. As we reckon, the combinatorial challenge of partial-labeling and long-tail learning lies in matching between a decent marginal prior distribution with drawing the pseudo labels. To cope with these problems, SoLar comprises two components. On one hand, SoLar formulates an optimal transport objective to facilitate the refinement of pseudo-labels to match the marginal class prior. The resulted constrained optimization objective encourages the model to properly draw the tail labels. On the other hand, the PLL setting manifests difficulty in straightforwardly estimating the class prior. SoLar henceforth incorporates two novel techniques—a moving-average estimation approach with a complementary reliable set sampling mechanism. *The pseudo-code is summarized in Algorithm 1.*

### 3.1 Sinkhorn Pseudo-Label Refinery

**Optimal Transport Objective.** We formalize an optimal transport problem for proper label assignments. At each training step, SoLar searches for pseudo-labels $\boldsymbol{Q}$ close to the current classifier's predictions $\boldsymbol{P}$, while subject to some constraints:

$$\min_{\boldsymbol{Q} \in \Delta} E(\boldsymbol{Q}, \boldsymbol{P}) = \langle \boldsymbol{Q}, -\log(\boldsymbol{P}) \rangle = -\sum_{i=1}^{n}\sum_{j=1}^{L} q_{ij} \log p_{ij} \tag{1}$$

$$\text{s.t. } \Delta = \{[q_{ij}]_{n \times L} | \boldsymbol{Q}^{\top}\boldsymbol{1}_n = \boldsymbol{r}, \boldsymbol{Q}\boldsymbol{1}_L = \boldsymbol{c}, q_{ij} = 0 \text{ if } j \notin S_i\} \subset [0,1]^{n \times L}.$$

$\boldsymbol{r}$ is an $L$-dimensional probability simplex that indicates the prior class distribution. In other words, we would like the summation of the $j$-th column in $\boldsymbol{Q}$ (*i.e.*, the total probability mass predicted as

class $j$) to match the prior class probability. Note that, here we temporarily assume we have a decent estimation of the class priors and we will describe the means of estimation in the following Section 3.2. The column vector $c = \frac{1}{n}\mathbf{1}_n$ indicates that our $n$ training examples are sampled uniformly. In Eq. (1), with a slight abuse of notation, $Q$ is essentially a joint probability matrix post to the refinery. That is, each $x_i$ is treated as a supplier holding $\frac{1}{n}$ units of goods while the $j$-th label needs $q_{ij}$ units of goods. To obtain pseudo-labels, we take a re-scaled solution $nQ$ bearing an assumption of a uniform marginal $\frac{1}{n}$ of $x_i$.

With further scrutiny of the objective function above, this formulation complies with two constraints we posited, such that: (i)-it avoids biased assignment towards (only) the head labels in a long-tailed PLL setup, with $Q^\top \mathbf{1}_n = r$ constraining the distribution of pseudo-labeling to match the prior of the class labels. (ii)-the probability mass of $Q$ in each row is driven to be distributed within the candidate label set of a sample $x_i$.

In Eq. (1), we use $-\log(p_{ij})$ as cost such that predictions with higher confidence enjoy a lower cost. This can also be replaced by other cost functions $h(p_{ij})$ as long as $h$ is monotonically decreasing. But, we show that the negative log-likelihood cost results in a statistically consistent risk estimator.

**Theorem 1.** *Denote $\tilde{\Delta}$ the set of measures on $\mathcal{X} \times \mathcal{Y}$ that satisfies the constraints in Eq. (1). Let $\mathcal{D}_p$ be the distribution of tuple $(x, S)$. The population risk of our method is as follows,*

$$\mathcal{R}(f) = \inf_{\varpi \in \tilde{\Delta}} \mathbb{E}_{(x,S) \sim \mathcal{D}_p} KL(f(x) \| \varpi(y|x)). \tag{2}$$

*Let $f^{**}$ and $f^*$ denote the optimal classifiers trained on fully-supervised data with cross-entropy loss and PLL data with $\mathcal{R}(f)$, respectively. Assume the small ambiguity condition [5] is satisfied. Then, under the deterministic scenario and ignoring the null set, we have $f^* = f^{**}$.*

We refer the readers to Appendix A.1 for the proof. Theorem 1 indicates that the optimal classifier $f^{**}$ can be recovered by our algorithm at a population level. Compared with other theoretically consistent PLL risks [6, 16], our risk preserves the marginal class distribution of the estimated pseudo-labels, and is reasonably more favorable in the long-tailed learning setup.

**Fast Approximation.** To resolve Eq. (1), we adapt the well-known Sinkhorn-Knopp algorithm [13] for efficient optimization. Formally, we define a matrix $M$ such that $m_{ij} = p_{ij}^\lambda \mathbb{I}(j \in S_i)$, where $\lambda > 0$ is a smoothing regularization coefficient. The pseudo-labels are obtained by,

$$Q = n \cdot \text{diag}(\alpha) M \text{diag}(\beta),$$
$$\text{with iteratively updated} \quad \alpha \leftarrow c./(M\beta), \quad \beta \leftarrow r./(M^\top \alpha). \tag{3}$$

Here ./ denotes element-wise division. $\alpha \in \mathbb{R}^n$ and $\beta \in \mathbb{R}^L$ are known as scaling coefficients vectors. Detailed derivation are provided in Appendix A.2. Here, we set $m_{ij} = 0$ if $j \notin S_i$, which is equivalent to define $\log(0) = -\inf$. Thus, $Q$ will also be assigned 0s outside the candidate sets.

Inspired by [17], we also involve a queue acceleration trick to avoid traveling the whole training set. We found that Eq. (3) typically converges to a satisfactory result in $< 50$ rounds. With only simple algebra operations, this can be efficiently implemented on GPUs; see Appendix C.3.

### 3.2 Iterative Class Prior Estimation

Unlike conventional long-tailed learning scenarios [14, 15], the PLL setup is presented with label ambiguity in the proximity of candidate sets. This makes it intractable to estimate the marginal class prior distribution, especially by counting training samples grouped by class labels.

In SoLar, towards estimating the class marginals, our aim is to propose a solution with (almost) full autonomy that can adapt to the data distribution and the training state of the model. Here, we propose two interdependent techniques to achieve this goal. The resulted estimation is utilized in Eq (1). The core of our method is two-fold: (i)-we propose a moving-average style updating rule for class prior estimation, resulting in *stable training dynamics*; (ii)-we carefully devise a sample selection mechanism for robust training, while *preserving the marginal label distribution* as much as possible.

**Moving-Average Distribution Updating.** We propose using the model predicted results as a proxy for class prior estimation. In spite of its simplicity, we cannot ignore the fact during the early stage of training, the predicted labels can be largely imprecise, not trustworthy, and biased towards

the head labels. Therefore, we propose an update mechanism in a moving-average fashion. Since we hold no assumption on the true class prior, the updating scheme is initiated from a uniform class prior $\boldsymbol{r} = [1/L, \ldots, 1/L]$. Respectively at each cycle (often defined by epochs), SoLar refines the distribution as follows.

$$\boldsymbol{r} \leftarrow \mu\boldsymbol{r} + (1-\mu)\boldsymbol{z}, \quad \text{where } z_j = \frac{1}{n}\sum_{i=1}^{n} \mathbb{I}(j = \arg\max_{j' \in S_i} f_{j'}(\boldsymbol{x}_i)), \tag{4}$$

where $\mu \in [0, 1]$ is a preset scalar. As displayed, the class prior is progressively updated alongside stabler training dynamics through a linear function. As the training proceeds, the model becomes more accurate and thus the estimated distribution becomes more reliable.

**Class-wise Reliable Sample Selection.** In the early stages of training, the class prior estimation can be inaccurate due to the poor pseudo label predictions. Thus, we posit that the lack of proper sample selection at early stages may jeopardize the marginal prior estimation. As common as it is for PLL learning dynamics, we want to reduce the variance of the pseudo-label quality, via a reliable sampling algorithm. This module works together with the moving-average class prior estimation.

A brute-force but effective strategy is to employ the small-loss sampling criterion in noisy-label learning [18], which assumes low-quality pseudo-labels can likely be characterized by a large loss. However, in a long-tailed setup, the tail labels can also incur a large loss because of their low-shot nature, making them rarely selected. We empirically show this phenomenon in Section 4.3. We hence implement the following procedures into the proposed SoLar framework. The goal of such a module is to make sure the coverage of the tail labels as well as to keep the marginal prior unbiased.

To begin with, to make sure that all labels — particularly the tail labels — are coverd, we split the batch of data $B$ into $L$ slices that are indexed by the *argmax* taken from the refined pseudo-labels: $B_j = \{(\boldsymbol{x}_i, \boldsymbol{q}_i) | j = \arg\max_{j' \in S_i} q_{ij'}\} \subset B$. In what follows, we derive two combinatorial rules on the selectivity of the training samples:

(i) Based on the aforementioned setup, we modify the small-loss criterion in the noisy-label learning [18] and adapt it to a long-tailed setup. Specifically, we first compute the instance-wise loss by marginalizing over the class index: $l_i = -\sum_{j=1}^{L} q_{ij}\log(p_{ij})$. Then, within the provided set $B_j$, we anchor the total sample number to be $k = \min(\lceil r_j\rho|B|\rceil, |B_j|)$, which is associated with $r_j$ for *distribution-preserving*. The samples are being selected with top-$k$ smallest $l_i$.

(ii) A parallel mechanism is inspired by the application of pseudo-labeling schemes in semi-supervised learning [19]. To adopt such approaches in SoLar, we purposely select those samples having a very high confidence score which is measured by the cosine similarity: $e_i = \boldsymbol{q}_i^\top \boldsymbol{p}_i$. As such, this second criteria selects samples that meet the inequality $e_i > \tau$. This mechanism further ensures high utility on the (likely) well-disambiguated samples.

Notice that, $\rho, \tau > 0$ are two thresholding hyper-parameters. The two introduced mechanisms are established to work collaboratively. Simply put, samples that satisfy either criterion are used for training the classifier. We will show later in Section 4.3 the efficacy of our sample selection mechanisms.

## 4 Experiments

In this section, we experimentally analyze the proposed SoLar method under various scenarios for the imbalanced PLL problem. More empirical results can be found in Appendix C.

### 4.1 Setup

**Datasets.** First, we evaluate SoLar on two long-tailed datasets CIFAR10-LT and CIFAR100-LT introduced in [20, 21]. The training images are randomly removed class-wise to follow a pre-defined imbalance ratio $\gamma = \frac{n_1}{n_L}$, where $n_j$ is the image number of the $j$-th class. For convenience, class indices are sorted based on the class-wise sample size, in descending order with $n_1 \geq \ldots \geq n_L$. We have $n_1 = 5,000$ for CIFAR10-LT and $n_1 = 500$ for CIFAR100-LT. We use different imbalance ratios to evaluate the performance of SoLar, with $\gamma \in \{50, 100, 200\}$ for CIFAR10-LT and $\gamma \in \{10, 20, 50\}$ for CIFAR100-LT. We then generate partially labeled datasets by manually flipping negative labels $\bar{y} \neq y$ to false-positive labels with probability $\phi = P(\bar{y} \in Y | \bar{y} \neq y)$, which follows

Table 1: Accuracy comparisons on CIFAR10-LT and CIFAR100-LT under various flipping probability $\phi$ and imbalance ratio $\gamma$. Bold indicates superior results.

| | CIFAR10-LT | | | | | |
|---|---|---|---|---|---|---|
| Methods | $\phi = 0.3$ | | | $\phi = 0.5$ | | |
| | $\gamma = 50$ | $\gamma = 100$ | $\gamma = 200$ | $\gamma = 50$ | $\gamma = 100$ | $\gamma = 200$ |
| MSE | $61.13_{\pm 1.08}$ | $52.59_{\pm 0.48}$ | $48.09_{\pm 0.45}$ | $49.61_{\pm 1.42}$ | $43.90_{\pm 0.77}$ | $39.52_{\pm 0.70}$ |
| EXP | $52.93_{\pm 3.44}$ | $43.59_{\pm 0.16}$ | $42.56_{\pm 0.44}$ | $50.62_{\pm 3.00}$ | $43.69_{\pm 2.72}$ | $41.07_{\pm 0.62}$ |
| LWS | $44.51_{\pm 0.03}$ | $43.60_{\pm 0.12}$ | $42.33_{\pm 0.58}$ | $24.62_{\pm 9.67}$ | $27.33_{\pm 1.84}$ | $28.74_{\pm 1.86}$ |
| VALEN | $58.34_{\pm 1.05}$ | $50.20_{\pm 6.55}$ | $46.98_{\pm 1.24}$ | $40.02_{\pm 1.88}$ | $37.10_{\pm 0.88}$ | $36.61_{\pm 0.57}$ |
| CC | $78.76_{\pm 0.27}$ | $71.86_{\pm 0.78}$ | $63.38_{\pm 0.79}$ | $73.09_{\pm 0.40}$ | $64.88_{\pm 1.03}$ | $54.41_{\pm 0.85}$ |
| PRODEN | $81.95_{\pm 0.19}$ | $71.09_{\pm 0.54}$ | $63.00_{\pm 0.54}$ | $66.00_{\pm 3.60}$ | $62.17_{\pm 3.36}$ | $54.65_{\pm 1.00}$ |
| PiCO | $75.42_{\pm 0.49}$ | $67.73_{\pm 0.64}$ | $61.12_{\pm 0.67}$ | $72.33_{\pm 0.08}$ | $63.25_{\pm 0.64}$ | $53.92_{\pm 1.64}$ |
| SoLar (ours) | $\mathbf{83.80}_{\pm 0.52}$ | $\mathbf{76.64}_{\pm 1.66}$ | $\mathbf{67.47}_{\pm 1.05}$ | $\mathbf{81.38}_{\pm 2.84}$ | $\mathbf{74.16}_{\pm 3.03}$ | $\mathbf{62.12}_{\pm 4.33}$ |

| | CIFAR100-LT | | | | | |
|---|---|---|---|---|---|---|
| Methods | $\phi = 0.05$ | | | $\phi = 0.1$ | | |
| | $\gamma = 10$ | $\gamma = 20$ | $\gamma = 50$ | $\gamma = 10$ | $\gamma = 20$ | $\gamma = 50$ |
| MSE | $49.92_{\pm 0.64}$ | $43.94_{\pm 0.86}$ | $37.77_{\pm 0.40}$ | $42.99_{\pm 0.47}$ | $37.19_{\pm 0.72}$ | $31.49_{\pm 0.35}$ |
| EXP | $25.86_{\pm 0.94}$ | $24.84_{\pm 0.40}$ | $23.58_{\pm 0.47}$ | $24.82_{\pm 1.41}$ | $21.27_{\pm 1.24}$ | $19.88_{\pm 0.43}$ |
| LWS | $48.85_{\pm 2.16}$ | $35.88_{\pm 1.29}$ | $19.22_{\pm 8.56}$ | $6.10_{\pm 2.05}$ | $7.16_{\pm 2.03}$ | $5.15_{\pm 0.36}$ |
| VALEN | $49.12_{\pm 0.58}$ | $42.05_{\pm 1.52}$ | $35.62_{\pm 0.43}$ | $33.39_{\pm 0.65}$ | $30.67_{\pm 0.11}$ | $24.93_{\pm 0.87}$ |
| CC | $60.36_{\pm 0.52}$ | $54.33_{\pm 0.21}$ | $45.83_{\pm 0.31}$ | $57.91_{\pm 0.41}$ | $51.09_{\pm 0.48}$ | $\mathbf{41.74}_{\pm 0.41}$ |
| PRODEN | $60.31_{\pm 0.50}$ | $50.39_{\pm 0.96}$ | $42.29_{\pm 0.44}$ | $47.32_{\pm 0.60}$ | $41.82_{\pm 0.55}$ | $35.11_{\pm 0.08}$ |
| PiCO | $54.05_{\pm 0.37}$ | $46.93_{\pm 0.65}$ | $38.74_{\pm 0.11}$ | $46.49_{\pm 0.46}$ | $39.80_{\pm 0.34}$ | $34.97_{\pm 0.09}$ |
| SoLar (ours) | $\mathbf{64.75}_{\pm 0.71}$ | $\mathbf{56.47}_{\pm 0.76}$ | $\mathbf{46.18}_{\pm 0.85}$ | $\mathbf{61.82}_{\pm 0.71}$ | $\mathbf{53.03}_{\pm 0.56}$ | $40.96_{\pm 1.01}$ |

the settings in previous works [6, 22]. The final candidate label set is composed of the ground-truth label and the flipped false-positive labels. We choose $\phi \in \{0.3, 0.5\}$ for CIFAR10-LT and $\phi \in \{0.05, 0.1\}$ for CIFAR100-LT. For all experiments, we report the mean and standard deviation based on 3 independent runs (with different random seeds).

**Baselines.** We compare SoLar with six state-of-the-art partial-label learning methods: 1) PiCO [7] leverages contrastive learning to disambiguate the candidate labels by updating the pseudo-labels with contrastive prototype labels. 2) PRODEN [6] is also a pseudo-labeling method that iteratively updates the latent label distribution by re-normalized classifier prediction. 3) VALEN [23] assumes the candidate labels are instance-dependent and recovers the latent label distributions by a Bayesian parametrization model. 4) LWS [22] also works in a pseudo-labeling style, which weights the risk function by considering the trade-off between losses on candidate labels and non-candidate ones. 5) CC [16] is a classifier-consistent method that assumes the candidate label set is uniformly sampled. 6) MSE and EXP [24] utilize mean square error and exponential loss as the risk estimators. All the hyper-parameters are searched according to the original papers.

**Implementation details.** We use an 18-layer ResNet as the feature backbone. The model is trained for 1000 epochs using a standard SGD optimizer with a momentum of 0.9. The initial learning rate is set as 0.01, and decays by the cosine learning rate schedule. The batch size is 256. These configurations are applied for SoLar and all baselines for fair comparisons. For SoLar, we devise a pre-estimation training stage, where we run a model for $100/20$ epochs (on CIFAR10/100-LT) respectively to obtain a coarse-grained class prior. After that, we re-initialize the model weights and run with this class prior. For our Sinkhorn-Knopp algorithm, we fix the smoothing regularization parameter as $\lambda = 3$ and the length of the queue for acceleration as $64$ times batch size. The moving-average parameter $\mu$ for class prior estimation is set as $0.1/0.05$ in the first stage and fixed as $0.01$ later. For class-wise reliable sample selection, we linearly ramp up $\rho$ from $0.2$ to $0.5/0.6$ in the first 50 epochs and fix the high-confidence selection threshold $\tau$ as $0.99$. To improve the representation ability of SoLar, we further involve consistency regularization [19] and Mixup [25] techniques on reliable examples; see Appendix B.1 for more details. For fair comparisons, we equip all the baselines except PiCO with these two techniques as well.

Table 2: Different shots accuracy comparisons on CIFAR10-LT ($\phi = 0.5, \gamma = 100$) and CIFAR100-LT ($\phi = 0.1, \gamma = 20$). The best results are marked in bold and the second-best marked in underline.

| Methods | CIFAR10-LT | | | CIFAR100-LT | | |
|---|---|---|---|---|---|---|
| | Many | Medium | Few | Many | Medium | Few |
| MSE | 81.11 | 42.03 | 9.18 | 57.46 | 37.57 | 16.53 |
| EXP | 92.64 | 39.75 | 0.00 | 60.35 | 3.99 | 0.00 |
| LWS | 89.09 | 1.52 | 0.00 | 20.80 | 0.86 | 0.00 |
| VALEN | 85.30 | 28.78 | 0.00 | 58.74 | 16.25 | 0.07 |
| CC | 94.56 | 65.61 | 34.22 | 73.03 | 52.15 | 28.05 |
| PRODEN | **96.83** | 72.18 | 14.17 | **76.86** | 43.14 | 5.43 |
| PiCO | 93.33 | 66.14 | 29.30 | 70.75 | 42.42 | 6.14 |
| SoLar (ours) | 96.50 | **76.01** | **49.34** | 74.33 | **54.09** | **30.62** |

Table 3: Ablation results on CIFAR10-LT ($\phi = 0.5, \gamma = 100$) and CIFAR100-LT ($\phi = 0.1, \gamma = 20$).

| Ablation | CIFAR10-LT | | | | CIFAR100-LT | | | |
|---|---|---|---|---|---|---|---|---|
| | All | Many | Med. | Few | All | Many | Med. | Few |
| SoLar | 74.16 | 96.50 | 76.01 | 49.34 | 53.03 | 74.33 | 54.09 | 30.62 |
| PRODEN w S+CPE | 62.83 | 96.80 | 69.79 | 19.58 | 41.69 | 75.70 | 42.29 | 7.06 |
| SoLar w/o Pre-Est | 63.97 | 79.97 | 70.21 | 39.67 | 50.51 | 68.09 | 54.15 | 29.18 |
| SoLar w Cand-Est | 74.34 | 96.37 | 74.47 | 52.14 | 52.78 | 76.15 | 53.93 | 28.22 |
| SoLar-Oracle | 74.11 | 96.81 | 77.10 | 47.43 | 54.16 | 77.91 | 56.94 | 27.55 |
| SoLar w/o S | 29.61 | 86.10 | 6.95 | 3.34 | 35.80 | 66.85 | 32.06 | 8.61 |
| SoLar w GS | 53.01 | 96.58 | 60.10 | 0.00 | 33.62 | 72.03 | 23.91 | 5.21 |
| SoLar w/o S-SL | 47.49 | 96.46 | 46.37 | 0.00 | 12.80 | 30.42 | 7.97 | 0.15 |

## 4.2 Main Results

**SoLar achieves SOTA results.** As shown in Table 1, SoLar significantly outperforms all the rivals by a notable margin under various settings of imbalance ratio $\gamma$ and label ambiguity degree $\phi$. Specifically, on CIFAR10-LT dataset with $\phi = 0.5$, we improve upon the best baseline by **8.29%**, **9.28%**, and **7.47%** when the imbalance ratio $\gamma$ is set to 50, 100, and 200 respectively. Moreover, most baselines display significant performance degradation as $\phi$ increases, whereas SoLar remains competitive. We also verify the effectiveness of SoLar on a more challenging CIFAR100-LT dataset with 10 times more classes and thus stronger label ambiguity. Notably, the performance gap between SoLar and baselines remains substantial. An interesting observation is that CC obtains impressive performance on the CIFAR-100 dataset, but is still inferior to SoLar in most cases. These observations clearly validate the superiority of SoLar.

**Results on different groups of labels.** We show that SoLar achieves overall strong performance on *both* head and tail classes. To see this, we report accuracy on three groups of classes with different sample sizes. Recall from Section 4.1 that the class indices are sorted based on the sample size, in descending order. We divide the classes into three groups: many ($\{1, 2, 3\}$), medium ($\{4, 5, 6, 7\}$), and few ($\{8, 9, 10\}$) shots for CIFAR10-LT and many ($\{1, \ldots, 33\}$), medium ($\{34, \ldots, 67\}$), and few ($\{68, \ldots, 100\}$) shots for CIFAR100-LT. Table 2 shows the accuracy of different groups on both CIFAR10-LT and CIFAR100-LT. On few-shot classes, the gaps between SoLar and the best baseline are **15.12%** and **2.57%** on CIFAR10-LT and CIFAR100-LT. Moreover, SoLar retains a competitive performance on many-shot labels (*i.e.*, head classes). In contrast, most baselines exhibit degenerated performance on tail classes, especially on the CIFAR100-LT dataset. The comparisons highlight that SoLar poses a good disambiguation ability in the long-tailed PLL setup.

## 4.3 Ablation Studies

In this section, we present our main ablation analysis to show the effectiveness of SoLar. The main results of our ablation studies are shown in Table 3; more results can be found in Appendix C.4.

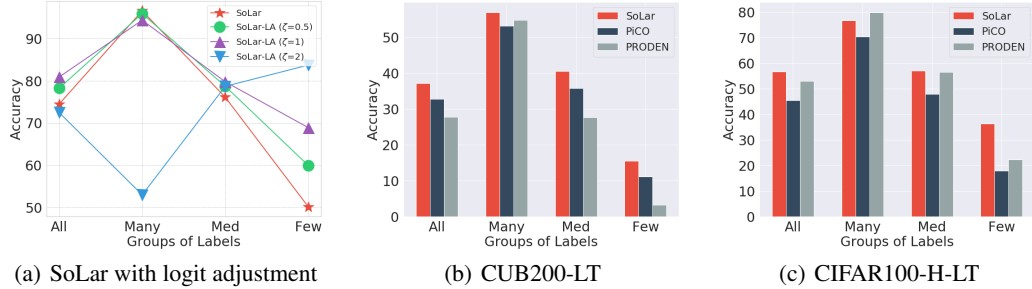

| (a) SoLar with logit adjustment | (b) CUB200-LT | (c) CIFAR100-H-LT |

Figure 2: (a) Performance comparisons of SoLar and SoLar with logit adjustment (SoLar-LA) on CIFAR10-LT ($\phi = 0.5, \gamma = 100$). $\zeta$ is the temperature parameter of logit adjustment. (b) Performance comparisons of SoLar and baselines on the fine-grained CUB200-LT dataset. (c) Performance comparisons of SoLar and baselines on the CIFAR100-LT dataset with hierarchical labels (CIFAR100-H-LT).

**Effect of Sinkhorn label refinery.** It is of interest to see whether the performance improvement comes from our Sinkhorn label refinery procedure. To see this, we equip PRODEN with our sample selection and class prior estimation mechanism (dubbed *PRODEN w S+CPE*). As shown in Table 3, PRODEN w S+CPE still fails to disambiguate labels on the tail, which validates the superiority of our Sinkhorn label refinery procedure.

**Effect of class prior estimation.** To verify the effectiveness of class prior estimation in SoLar, we compare with three variants: 1) *SoLar w/o Pre-Est* which removes the pre-estimation stage of SoLar and trains from uniform class prior; 2) *SoLar w Cand-Est* initializes the class prior distribution by counting per-class candidate numbers, *i.e.*, $r_j = \sum_{i=1}^{n} \mathbb{I}(j \in S_i)$; 3) *SoLar-Oracle* also removes the pre-estimation stage but trains from the real class prior. From Table 3, we draw three salient observations. First, SoLar outperforms SoLar w/o Pre-Est, which suggests that using a coarsely-estimated class prior is more beneficial than using a uniform one. Secondly, SoLar w Cand-Est obtains competitive results, showing an alternative way for initializing $r$. Finally, we contrast with SoLar-Oracle, which theoretically serves as an upper bound by using the real class prior. We observe that SoLar's performance favorably matches or even outperforms SoLar-Oracle on few-shot classes. The results are encouraging since SoLar does not require access to real class prior at all.

**Effect of sample selection.** Next, we experiment with different sample selection strategies, including: 1) *SoLar w/o S* which regards all examples as clean samples and does not perform selection; 2) *SoLar w GS* which performs selection in a global manner rather than class-wise; 3) *SoLar w/o S-SL* which only selects high-confidence examples. From Table 3, we can observe that SoLar w/o S worsened the performance because many pseudo-labels are unreliable at the beginning. SoLar w/o GS works relatively well on data-rich labels but suffers on the tail. This happens because the majority classes dominate the sample selection. SoLar w/o S-SL performs similarly to SoLar w/o GS on CIFAR10-LT, but diverges on CIFAR100-LT. In the beginning, few examples are selected as the classifier is less confident, resulting in degenerated solution. Our results indicate that our sample selection procedure is indispensable to SoLar in handling long-tailed class distribution.

## 4.4 Further Analysis

**Combining SoLar with logit adjustment.** While our work mainly focuses on label disambiguation from the PLL perspective, potential improvement is straightforward by applying long-tailed learning techniques to well-disambiguated datasets. To this end, we equip SoLar with *logit adjustment* [14] (dubbed *SoLar-LA*), a state-of-the-art LTL technique, for improved performance. Denote the logits of our model by $g(\cdot)$, *i.e.*, the output of the network prior to the softmax layer. Given a testing data $x$, SoLar-LA made prediction by $\arg\max_{j \in [L]} g_j(x) - \zeta \cdot \log(r_j)$, where $\zeta$ is a temperature parameter for calibration. This post-hoc corrected prediction is known to be theoretically consistent for minimizing the balanced error [14]. Note that logit adjustment cannot be applied to the original PLL data directly, as it requires the true class prior as well as a classifier trained by minimizing the long-tailed empirical risk. Fortunately, SoLar achieves both goals. Hence, we implement SoLar-LA by using the classifier as well as the estimated class prior of SoLar. Figure 2 (a) reports the performance of SoLar-LA with varying $\zeta$ values. With a proper $\zeta$, SoLar-LA improves upon SoLar

by **6.72**% on CIFAR10-LT. This suggests that SoLar is compatible with existing LTL methods and opens the door to exploring more advanced LTL techniques.

**Results on fine-grained partial-label learning.** In practice, semantically similar classes can lead to significant label ambiguity, as exemplified in Figure 1 (a). To test the limit of SoLar, we follow PiCO [7] and evaluate on two fine-grained datasets: 1) CUB200-LT [26] dataset with 200 bird species; 2) CIFAR100-LT with hierarchical labels (CIFAR100-H-LT), where the candidate labels are generated within the same superclass[2]. We set $\phi = 0.05, \gamma = 5$ for CUB200-LT and $\phi = 0.5, \gamma = 20$ for CIFAR100-H-LT. In Figure 2 (b) and (c), we compare SoLar with two strongest baselines PRODEN and PiCO, where SoLar improves the best baselines by a large margin (**4.37**% on CUB200-LT and **3.67**% on CIFAR100-H-LT). These results clearly validate the effectiveness of SoLar, even when the dataset presents severe label ambiguity.

**Results on real-world long-tailed learning data.** It is of interest to verify the effectiveness of SoLar on real-world imbalanced datasets. To this end, we conduct experiments on the large-scale SUN397 dataset [27] containing 108,754 RGB images and 397 scene classes; more results on real-world and imbalanced PLL datasets are shown in Appendix C.6. For the

Table 4: Performance comparisons on the SUN397 dataset with $\phi = 0.05$.

| Methods | All | Many | Medium | Few |
|---|---|---|---|---|
| PRODEN | 51.14 | 76.98 | 58.17 | 17.61 |
| PiCO | 29.54 | 57.91 | 21.27 | 9.33 |
| SoLar | **61.58** | **77.36** | **62.59** | **44.49** |

SUN397 dataset, we hold out 50 samples per class for testing and the resulting training set has an imbalanced ratio of $2311/50 \approx 46$. Similarly, we synthesize a partial-label dataset set with an ambiguity degree of $\phi = 0.05$; see Appendix C.1 for more details. From Table 4, we can observe that SoLar still outperforms the baselines by a substantial margin.

## 5  Related Work

**Partial-label Learning** (PLL) [4] allows each example to be equipped with a candidate label set with the ground-truth label being concealed. A plethora of PLL methods [5, 28, 29, 30, 31] have been developed. For example, maximum margin-based methods [32, 33] maximize the margins between candidate labels and the remaining ones. Graph-based approaches [34, 8, 9, 10, 35] typically leverage nearest neighbors to smooth the logical label vectors. Recently, pseudo-labeling methods [16, 6, 22, 36, 7, 37] become increasingly prevalent due to their promising performance, *e.g.*, PRODEN [6] re-normalizes classifier's outputs and PiCO [7] updates pseudo-targets by using contrastively learned prototypes. However, most of these works overlook the fact that class distribution can be imbalanced. Our empirical studies demonstrate that state-of-the-art PLL methods fail to disambiguate true labels on the minority classes, which motivates us to explore the optimal transport-based label disambiguation technique for PLL.

**Long-tailed Learning** (LTL) assumes that the training data follow Zipf's law that few labels pose a large number of instances, which is observed in a wide range of applications [11]. To cope with the LTL problem, sampling-based methods [38, 39] re-balance different classes at a data level, *e.g.*, up-sampling on minority classes. Re-weighting methods [20, 14, 40, 41, 15] typically adjust the weights of instances or classes that calibrates the classifier predictions to be balanced. Some works also explore transfer learning [42, 43] or ensemble learning [44, 45] in LTL. A recent trend in LTL [46, 47] is to decompose representation and classifier learning. Nevertheless, these methods assume a full-supervision of training data, which is unrealistic in many real-world applications. It has motivated researchers to study the LTL problem in weak-supervision scenarios, such as semi-supervised learning [48, 21] and noisy-label learning [49, 50, 51]. In this work, we explore a new formulation of LTL where only candidate labels are provided.

**Optimal Transport** (OT) [12] is originally proposed to depict the distance between two measures. Recently, it has drawn huge attention from different fields in machine learning, including semi-supervised learning [52], domain adaptation [53, 54], object detection [55], and generative models [56]. The most related one to our work is OTLM [15], which also applies OT to the supervised LTL problem. By assuming a known test class distribution, OTLM performs OT-based post-hoc correction on model predictions to obtain balanced results. In contrast, our work targets the long-tailed PLL problem and holds no assumption on class priors.

---

[2]The CIFAR100 dataset comprises 20 superclasses, with 5 classes in each superclass.

# 6    Conclusion

In this work, we present a novel Sinkhorn label refinery framework (SoLar) for a challenging imbalanced partial-label learning problem. Key to our method, we derive an optimal transport objective that refines the pseudo-labels to match the true class prior. Additionally, we tackle the non-trivial class prior estimation problem by a moving-average technique, which is further guarded by a sample selection mechanism for controllable model training. Empirically, SoLar can run without knowledge of the real class distribution. Comprehensive experiments show that SoLar improves baseline algorithms by a significant margin. As we collect the real-world applications, the partial-labeled tasks often arise together with a long-tailed class distribution. That is, from the perspective of human annotators, a long-tailed label is much harder to tag and that may subsequently lead to a partial-labeled candidate set. In the future, we hope to explore this line of research in broader tasks. We hope our work will draw more attention from the community towards a broader view of tackling the imbalanced partial-label learning problem.

## Acknowledgments and Disclosure of Funding

This work was largely supported by the Key R&D Program of Zhejiang Province (Grant No. 2020C01024). JZ was supported by the Fundamental Research Funds for the Central Universities (No. 226-2022-00028). HW, MX were partially supported by the Key Research and Development Program of Zhejiang Province of China (No. 2021C01009) and Fundamental Research Funds for the Central Universities. LF was supported by the National Natural Science Foundation of China (Grant No. 62106028), Chongqing Overseas Chinese Entrepreneurship and Innovation Support Program.

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
