# SoLar: Sinkhorn Label Refinery for Imbalanced Partial-Label Learning
# (Appendix)

**Haobo Wang**[1]    **Mingxuan Xia**[2]    **Yixuan Li**[3]    **Yuren Mao**[2]
**Lei Feng**[45]    **Gang Chen**[1]    **Junbo Zhao**[1*]
[1]Key Lab of Intelligent Computing based Big Data of Zhejiang Province, Zhejiang University
[2]School of Software Technology, Zhejiang University
[3]Department of Computer Sciences, University of Wisconsin-Madison
[4]College of Computer Science, Chongqing University
[5]Center for Advanced Intelligence Project, RIKEN
{wanghaobo,xiamingxuan,yuren.mao,cg,j.zhao}@zju.edu.cn
sharonli@cs.wisc.edu, lfeng@cqu.edu.cn

# Appendix

## A   Theoretical Proofs

### A.1   Proof of Theorem 1

First, we provide the following lemma to show the consistency of the standard cross-entropy loss.

**Lemma 1.** *If the cross-entropy loss is used as loss function, the optimal classifier $f^{**}$ that minimizes the population risk $R_{CE}(f) = \mathbb{E}_{\boldsymbol{x},y}[-\log(f_y(x))]$ satisfies $f_y^{**}(\boldsymbol{x}) = p(y = i|\boldsymbol{x})$.*

*Proof.* We provide a proof sketch in the sequel, and a similar result has been shown in [1, 2]. The cross-entropy loss leads to the following optimization problem,

$$\min_f -\sum_{i=1}^{L} p(y = i|\boldsymbol{x}) \log(f_i(\boldsymbol{x})) \quad \text{s.t.} \quad \sum_{i=1}^{L} f_i(\boldsymbol{x}) = 1. \tag{1}$$

By introducing a multiplier $\xi$, the corresponding Lagrangian is as follows,

$$\mathcal{L}(f, \xi) = -\sum_{i=1}^{L} p(y = i|\boldsymbol{x}) \log(f_i(\boldsymbol{x})) + \xi(\sum_{i=1}^{L} f_i(\boldsymbol{x}) - 1). \tag{2}$$

Setting the derivative to 0 yields that,

$$f_i^{**}(\boldsymbol{x}) = \frac{1}{\xi} p(y = i|\boldsymbol{x}), \quad \sum_{i=1}^{L} f_i^{**}(\boldsymbol{x}) = \frac{1}{\xi} \sum_{i=1}^{L} p(y = i|\boldsymbol{x}) = 1. \tag{3}$$

We conclude that $\xi = 1$ and $f_y^{**}(\boldsymbol{x}) = p(y = i|\boldsymbol{x})$ holds.    □

Next, we provide the definition of the (theoretical) ambiguity degree [3] to establish the learnability of the PLL problem.

---

*Corresponding author.

36th Conference on Neural Information Processing Systems (NeurIPS 2022).

**Definition 1.** *Denote the distribution of triplet $(\boldsymbol{x}, y, S)$ by $\hat{\mathcal{D}}_p$, We define the ambiguity degree as,*

$$\kappa = \sup_{(\boldsymbol{x},y,S)\sim\hat{\mathcal{D}}_p, p(\boldsymbol{x},y)>0, \bar{y}\neq y} P(\bar{y} \in S|\boldsymbol{x}, y). \tag{4}$$

We say a data distribution satisfies the *small ambiguity degree condition* if $\kappa < 1$. This is a natural requirement. If $\kappa = 1$, there exists at least one label pair that always co-occur, and thus, it is impossible to find the optimal hypothesis given partial-labeled data.

Now, we provide the main proof sketch for Theorem 1. Note that we always seek an optimal joint probability matrix before model training, which is mainly designed for empirical measures of the data samples. At a population level, we aim to search for an optimal probability measure that meets the marginal constraints and candidate constraints. Note that we aim to minimize $E(\boldsymbol{Q}, \boldsymbol{P}) = \mathrm{KL}(\boldsymbol{Q}||\boldsymbol{P}) + H(\boldsymbol{Q})$ where $\mathrm{KL}(\cdot)$ is the Kullback–Leibler divergence and $H(\cdot)$ is an entropy regularizer. This gives rise to our population risk,

$$\mathcal{R}(f) = \inf_{\varpi\in\tilde{\Delta}} \mathbb{E}_{(\boldsymbol{x},S)\sim\mathcal{D}_p}\mathrm{KL}(f(\boldsymbol{x})||\varpi(y|\boldsymbol{x})). \tag{5}$$

Here we omit the entropy term $H(\varpi)$ as it serves as a regularizer. As the training labels are categorical, we may assume $H(p(y|x)) = 0$, and thus, the infimum still holds. Otherwise, we can offset it by setting the smoothing parameter $\lambda = 1$ in the Sinkhorn-Knopp approximation. When the pseudo-labels are fixed, this objective is exactly the cross-entropy loss whose target measure is $\varpi$.

To show the consistency, we first prove that $f^{**}$ is the minimizer of $\mathcal{R}(f)$. Substituting $f^{**}$ into $\mathcal{R}(f)$ and combining with Lemma 1, we obtain that,

$$\mathcal{R}(f^{**}) = \min_{\varpi\in\tilde{\Delta}} E_{(\boldsymbol{x},S)\sim\mathcal{D}_p}\mathrm{KL}(p(y|\boldsymbol{x})||\varpi(y|\boldsymbol{x})) = 0. \tag{6}$$

It is obvious that $\varpi = p(y|\boldsymbol{x})$ leads to the minimal KL-divergence.

On the other hand, we show that $f^{**}$ is the unique solution. We assume there exists another hypothesis $f'$ that minimizes Eq. (5) and holds a different prediction $y' \neq y$ from $f$ on at least one instance $(\boldsymbol{x}, y, S)$. By the deterministic assumption, we have $\mathcal{R}(f') = 0$, which holds only if $y'$ is invariably included in the candidate set $S$, *i.e.*, $P_{\hat{\mathcal{D}}_p}(y' \in S|\boldsymbol{x}, y) = 1$. Clearly, it violates the small ambiguity degree condition and causes a contradiction. Ignoring all the null set where $p(\boldsymbol{x}, y) = 0$, we conclude that the minimizer $f^*$ of our population risk equals the fully-supervised $f^{**}$.

The above discussion indicates that our method poses good performance guarantees like existing PLL methods [4, 2]. A similar loss $\mathcal{R}(f) = \mathbb{E}_{(\boldsymbol{x},S)\sim\mathcal{D}_p} \min_{y\in S} -\log(f_y(\boldsymbol{x}))$ is also discussed in [4], which is known as *minimal loss*. Its main difference from our objective is that we seek the best probability measure instead of an example-wise minimum. Our risk preserves the marginal constraint of the estimated pseudo-labels and is reasonably more favorable in the long-tailed learning setup.

## A.2 Derivation of the Sinkhorn-Knopp Iteration

In this section, we briefly introduce the derivation of the Sinkhorn-Knopp algorithm to ensure the integrity of our work.

Recall that Eq. (1) is a standard linear programming (LP) problem, and can be solved in polynomial time. But, considering the high volume of data points as well as potential large class space, common LP solvers typically become time-consuming. To this end, we investigate a smoothed version of this optimization problem for fast approximation. Technically, we further add a negative entropy regularization term to obtain the following objective,

$$\min_{\boldsymbol{Q}\in\Delta} \langle \boldsymbol{Q}, -\log(\boldsymbol{P}) \rangle - \frac{1}{\lambda}H(\boldsymbol{Q}). \tag{7}$$

This entropy regularizer is derived from an optimization perspective and is different from the one in our derivation of $E(\boldsymbol{Q}, \boldsymbol{P}) = \mathrm{KL}(\boldsymbol{Q}||\boldsymbol{P}) + H(\boldsymbol{Q})$. The resulting objective becomes smoothing and convex, and thus, can be efficiently solved.

It is worth pointing out that in the PLL problem, we have a special constraint that no probability mass is assigned outside the candidate label sets. To avoid introducing another set of annoying Lagrange

multipliers, we transform this constraint to infinity costs on non-candidate labels. That is, we define a cost matrix $T$ such that $T_{ij} = -\log(p_{ij})\mathbb{I}(j \in S_i) + \inf \mathbb{I}(j \notin S_i)$ and here we assume $0 \cdot \inf = 0$. Now, we obtain the Lagrangian of the new optimization problem,

$$\mathcal{L}(Q, u, v) = \langle Q, T \rangle - \frac{1}{\lambda}H(Q) + u^\top(Q\mathbf{1}_L - c) + v^\top(Q^\top\mathbf{1}_n - r), \tag{8}$$

where $u, v$ are Lagrange multipliers. As the original optimization is convex, the solution has to satisfy the Karush-Khun-Tucker (KKT) conditions. Therefore, we have the following equations,

$$\frac{\partial \mathcal{L}(Q, u, v)}{q_{ij}} = T_{ij} + \frac{1}{\lambda}(\log(q_{ij}) + 1) + u_i + v_j = 0. \tag{9}$$

Let $M$ be a matrix such that $m_{ij} = e^{-\lambda T_{ij}} = p_{ij}^\lambda \mathbb{I}(j \in S_i)$. We can get that,

$$q_{ij} = e^{-\frac{1}{2} - \lambda u_i} m_{ij} e^{-\frac{1}{2} - \lambda v_j}. \tag{10}$$

Thereby, solving the primal problem in Eq. (7) equals to finding the multipliers $u$ and $v$. Again, this is equivalent to get another two vectors $\alpha \in \mathbb{R}^n, \beta \in \mathbb{R}^L$ such that $\alpha_i = e^{-\frac{1}{2} - \lambda u_i}$ and $\beta_j = e^{-\frac{1}{2} - \lambda v_j}$, which are also known as scaling coefficients vectors. Then, we can get the following equation,

$$Q = \text{diag}(\alpha)M\text{diag}(\beta). \tag{11}$$

Recall that we enforce $Q$ to meet the following constraints,

$$Q\mathbf{1}_L = \text{diag}(\alpha)M\beta = c,$$
$$Q^\top\mathbf{1}_n = \text{diag}(\beta)M^\top\alpha = r. \tag{12}$$

This gives rise to an alternative coordinate descent algorithm for updating the scaling coefficients,

$$\alpha \leftarrow c./(M\beta), \quad \beta \leftarrow r./(M^\top\alpha). \tag{13}$$

It is also known as the Sinkhorn-Knopp fixed point iteration. We refer the readers to [5] for its convergence properties. Empirically, we find that setting $\lambda = 3$ and running 50 steps are enough to get a satisfactory solution. The final step is to take a re-scaled the minimizer $nQ$ in Eq. (11), since $Q$ serves as a joint probability matrix and the posterior is calculated by $\frac{p(x,y)}{p(x)} = np(x, y)$. Without loss of generality, we slightly abuse the notation $Q$ to denote the obtained pseudo-labels.

The core of our algorithm is to replace our zero constraints on non-candidates with infinity cost, which equals defining $\log(0) = -\inf$. These infinity values are then mapped back to 0s in the Sinkhorn-Knopp iteration, ensuring the feasibility of the calculation.

# B   Practical Implementation

In this section, we describe several details of the practical implementation of SoLar.

## B.1   Details of Representation Enhancement

A recent work PiCO [6] has shown that existing PLL methods are typically trapped in a *disambiguation-representation dilemma*, where the low-quality representation and imperfect label disambiguation mutually deteriorate each other. It can be even worse in the imbalanced setup because of low-shot examples on the tail. While PiCO pioneers the contrastive learning technique in PLL for superior performance, we show this can be achieved by a much simpler design. In contrast to the complicated network architecture in PiCO [6], we involve consistency training regularizer along with Mixup augmentation to improve the representation quality.

**Consistency Regularization.**   Recently, consistency regularization (CR) has been widely applied in weakly-supervised learning [7], which assumes that a classifier should produce similar class probability for a sample and its local augmented copies. Motivated by this, we also incorporated CR into PLL. Given an image $x_i$, we adopt two data augmentation modules SimAugment [8] and RandAugment [9] to obtain a weakly augmented image $x_i^w$ and strongly augmented counterpart $x_i^s$. During training, the weak one is utilized to produce pseudo-labels by our Sinkhorn label

refinery procedure as well as selecting the reliable subset. Then, we define the the CR loss $l_{\text{cr}} = -\sum_{j=1}^{L} q_{ij} \log(f_j(\boldsymbol{x}_i^s))$ that is the cross-entropy loss on $\boldsymbol{x}_i^s$ and $\boldsymbol{q}_i$.

**Mixup.** We further incorporate mixup training for improved performance. Given a pair of weakly-augmented examples $\boldsymbol{x}_i^w$ and $\boldsymbol{x}_j^w$ in the reliable set, we create a virtual training example by linearly interpolating both,

$$
\begin{aligned}
\boldsymbol{x}^m &= \sigma \boldsymbol{x}_i^w + (1 - \sigma)\boldsymbol{x}_j^w, \\
\boldsymbol{q}^m &= \sigma \boldsymbol{q}_i^w + (1 - \sigma)\boldsymbol{q}_j^w,
\end{aligned}
\tag{14}
$$

where $\sigma \sim \text{Beta}(\varsigma, \varsigma)$ and we simply set $\varsigma = 4$ without further tuning. Similarly, we define the mixup loss $l_{\text{mix}}$ as the cross-entropy loss on $\boldsymbol{x}^m$ and $\boldsymbol{q}^m$.

In our implementation, we add CR and Mixup on reliable examples only. For the remaining examples, since their pseudo-labels are unreliable, we train them with the classical re-normalized PLL loss $l_{\text{rn}}$ [4]. The final loss is defined by,

$$
l_{\text{cls}} = \eta(l_{\text{ce}} + l_{\text{cr}} + l_{\text{mix}}) + (1 - \eta)l_{\text{rn}},
\tag{15}
$$

where $l_{\text{ce}}$ is the original classification loss of weakly-augmented instances. We linearly ramp up $\eta$ from 0 to 0.9 in the first 50 epochs, which helps warm up the classifiers.

### B.2 Relaxed Solution for the Sinkhorn-Knopp Algorithm

Recall that we start from a uniform class prior. Empirically, most of our experiments run very well. But, in some cases, it can result in an unsolvable optimal transport objective. For example, if there are only a few examples holding one specific label as a candidate ($\ll \frac{n}{L}$), we have no hope to constrain the sum of pseudo-labels on this label to be the average number of instances. Thus, the Sinkhorn-Knopp iteration diverges.

In our implementation, once divergence occurs, we return a relaxed solution to the optimal transport problem. Concretely, we modify the matrix $\boldsymbol{M}$ by adding a small number $\epsilon = 1e^{-5}$ on its zero entries. In other words, we regard all negative labels as potential candidates but assign them very large costs. Now, $\boldsymbol{M}$ is an element-wise positive matrix, and the Sinkhorn fixed iteration process guarantees to converge. We can re-run the Sinkhorn-Knopp iteration to obtain a relaxed solution. Finally, we set the non-candidate labels as zero again, as they should not be assigned any probability mass. By then, our algorithm can safely run with a uniform distribution. This procedure is easy to implement and increases only $1\times$ computation at most. Moreover, as the class prior is estimated better and better, we typically do not need this relaxed solution anymore.

In practice, it is not likely that we are fully unknowledgeable of the class distribution. Thus, we can also use a good initialization to avoid the aforesaid problem. Empirically, we find that initializing with the ratio of candidate label number is also a good choice for SoLar.

## C  Additional Experimental Results

In this section, we report the additional empirical results of our proposed SoLar framework. All experiments are conducted on a workstation with 8 NVIDIA A6000 GPUs. The licenses of our employed datasets are unknown (non-commercial).

### C.1  Experimental Setups on Fine-grained Datasets and SUN397

In the sequel, we show the full experimental setups on fine-grained classification datasets. In particular, on CUB200-LT, we set the batch size as 128, and the length of the queue for Sinkhorn acceleration as 8 times batch size. We train the model for 500 epochs without the pre-estimation training stage. The ratio parameter $\rho$ ramps up from 0.2 to 0.5 in the first 50 epochs. Other hyper-parameters are the same as our default setting. On CIFAR100-H-LT, we simply adopt the default parameter configurations. The baselines are also fine-tuned to achieve their best results.

For the SUN397 dataset, we set the batch size as 128, and the queue length for Sinkhorn acceleration as 16 times batch size. We train the model for 20/200 epochs for distribution estimation and regular training. The gamma value is set as 0.1 at the pre-estimation stage. Other parameters are the same as

Table 5: Full results on the SUN397 dataset.

| Methods | SoLar ($\phi = 0.05$) | | | | SoLar ($\phi = 0.1$) | | | |
|---|---|---|---|---|---|---|---|---|
| | All | Many | Medium | Few | All | Many | Medium | Few |
| PRODEN | 51.14 | 76.98 | 58.17 | 17.61 | 35.96 | 76.62 | 29.48 | 1.40 |
| PiCO | 29.54 | 57.91 | 21.27 | 9.33 | 12.22 | 24.17 | 9.23 | 3.18 |
| SoLar | **61.58** | **77.36** | **62.59** | **44.49** | **55.64** | **76.78** | **57.87** | **31.86** |

Table 6: Accuracy comparisons of SoLar and SoLar with logits adjustment (SoLar-LA). The best results are marked in bold and the second-best marked in underline.

| Methods | CIFAR10-LT ($\phi = 0.5, \gamma = 100$) | | | | CIFAR100-LT ($\phi = 0.1, \gamma = 20$) | | | |
|---|---|---|---|---|---|---|---|---|
| | All | Many | Medium | Few | All | Many | Medium | Few |
| SoLar | 74.16 | **96.50** | 76.01 | 50.16 | 53.03 | **74.33** | 54.09 | 30.62 |
| SoLar-LA ($\zeta = 0.5$) | 78.23 | 95.91 | 78.70 | 59.94 | 54.17 | 73.05 | 55.75 | 33.66 |
| SoLar-LA ($\zeta = 1$) | **80.88** | 94.35 | **79.76** | 68.90 | **54.82** | 71.08 | 56.90 | 36.40 |
| SoLar-LA ($\zeta = 2$) | 72.48 | 52.98 | 78.68 | **83.71** | 54.37 | 64.45 | **57.70** | **40.86** |

the CUB200-LT dataset. As the SUN397 has a much larger scale, we calculate the empirical label distribution $z$ by recording batch-wise statistics during training. We find this on-the-fly counting strategy works as well as the default setup but is much faster. As reported in Table 5, SoLar retains substantial performance advantages in different ambiguity degrees.

## C.2 Full Results of SoLar with Logit Adjustment

In Table 6, we report the full results of SoLar-LA with varying $\zeta$ values on CIFAR10-LT and CIFAR100-LT. On CIFAR100-LT, SoLar-LA still outperforms SoLar with a proper $\zeta$. The results demonstrate that SoLar achieves promising disambiguation ability in the imbalanced PLL setup. Given well-disambiguated data, SoLar makes it possible to apply off-the-shelf LTL methods for further improvements.

## C.3 Running Time of the Sinkhorn-Knopp Algorithm

As we mentioned in Section 3.1, our Sinkhorn-Knopp algorithm can be efficiently implemented on GPUs. In Table 7, we report the total running time (1000 epochs) of the Sinkhorn-Knopp iterations as well as model training based on our implementation. We run SoLar with our default parameter configurations and evaluate using one NVIDIA A6000 GPU. During training, we main-

Table 7: Total running time (in hours) of the Sinkhorn-Knopp iterations and model training on CIFAR10-LT ($\gamma = 100$) and CIFAR100-LT ($\gamma = 20$).

| Dataset | CIFAR10-LT | CIFAR100-LT |
|---|---|---|
| Sinkhorn-Knopp | 0.08 | 0.10 |
| Model Training | 1.81 | 2.33 |

tain a queue of size $64 \times 256$ to store classifier predictions in previous 64 steps. Then, we concatenate the prediction in the current batch with the queue to run the Sinkhorn-Knopp algorithm. It can be shown that the Sinkhorn-Knopp iterations take less than $1/10$ time cost than regular model training (including forward pass and backward propagation). These results clearly validate the efficiency of our algorithm.

## C.4 More Ablation

**Effect of selection threshold $\rho$ and $\tau$.** We further investigate the effect of the small-loss selection ratio parameter $\rho$ and the high-confidence threshold $\tau$. Figure 3 (a) shows the performance of SoLar with varying $\rho$ on CIFAR10-LT (without ramp-up). When $\rho = 0$, SoLar simply selects high-confidence samples, which leads to an unsatisfactory performance on few-shot labels. The performance becomes much better as $\rho$ becomes larger and achieves the best when $\rho = 0.4$. But, when $\rho$ becomes too large, the model tends to overfit unreliable labels. Empirically, we find that SoLar works well in a wide range of $\rho$ and $\rho \approx 0.5$ is a good choice.

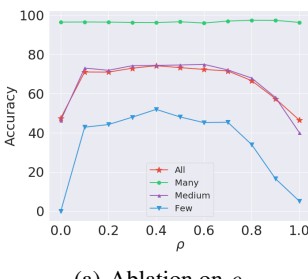
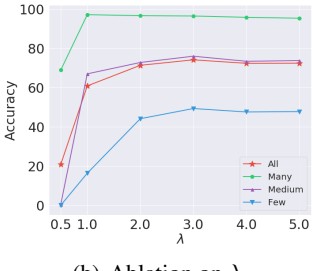
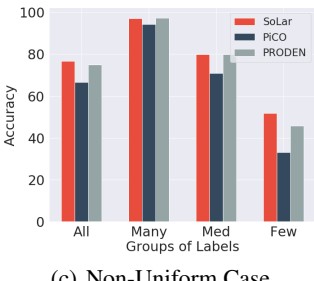

| (a) Ablation on $\rho$ | (b) Ablation on $\lambda$ | (c) Non-Uniform Case |
|---|---|---|

Figure 3: (a) Performance of SoLar with varying $\rho$ on CIFAR10-LT ($\phi = 0.5, \gamma = 100$). (b) Performance of SoLar with varying $\lambda$ on CIFAR10-LT ($\phi = 0.5, \gamma = 100$). (c) Performance comparisons of SoLar and baselines on CIFAR10-LT with non-uniform generated candidate labels.

Table 8 lists the result of SoLar with varying $\tau$ values. With a relatively small $\tau$, SoLar collects too many unconfident examples, which typically hurts the robustness of model training. When $\tau = 0.99$, SoLar achieves rather promising results on different groups of labels. A surprising observation is SoLar without high-confidence selection, *i.e.*, $\tau = 1$, achieves better results than SoLar on few-shot labels. We find the reason is that those well-disambiguated examples on the majority classes are overlooked, which poses an effect of down-sampling and benefits learning on the minority classes. As a negative effect, this sacrifices the performance of many-shot labels, which leads to an interesting trade-off. We empirically set $\tau = 0.99$ to retain relatively good performance on many-shot labels.

**Effect of Sinkhorn smooth parameter $\lambda$.** Next, we study the effect of the Sinkhorn smooth parameter $\lambda$. The results are shown in Figure 3 (b). In general, a too small $\lambda$ results in poor label assignment as the optimal transport objective becomes hard to be resolved. With a too large $\lambda$, our objective is over-smoothed and thus the resultant solution may deviate from the true one, which slightly degrades the performance as well. We empirically found that $\lambda = 3$ is a proper choice.

Table 8: Performance of SoLar with varying $\tau$ on CIFAR10-LT ($\phi = 0.5, \gamma = 100$).

| $\tau$ Values | All | Many | Medium | Few |
|---|---|---|---|---|
| 0.9 | 64.31 | 96.32 | 65.96 | 30.10 |
| 0.95 | 69.00 | 96.38 | 70.64 | 39.76 |
| 0.99 | 74.16 | **96.50** | **76.01** | 49.34 |
| 1.0 | **74.49** | 93.72 | 75.08 | **54.46** |

**The role of unreliable examples.** In our implementation, we enable unreliable examples to be trained with re-normalized PLL loss since their pseudo-labels can be noisy. This not only improves data utility but also serves as a warm-up mechanism for SoLar as we aim to train SoLar without knowledge of the true class prior. To see the role of unreliable examples, we also

Table 9: Performance of SoLar without training with unreliable examples.

| Class Prior | All | Many | Medium | Few |
|---|---|---|---|---|
| Oracle | 71.75 | 96.00 | 76.65 | 40.97 |
| Estimated | 61.85 | 68.00 | 72.17 | 41.94 |

evaluate SoLar without $l_{\mathrm{rn}}$ and the results are shown in Table 9. We observe that SoLar w/o $l_{\mathrm{rn}}$ obtains favorable results on tail labels, even trained with only reliable examples. This verifies the importance of our distribution-preserving sample selection mechanism. Given the oracle class prior, SoLar w/o $l_{\mathrm{rn}}$ obtains competitive performance and matches our main results. In practice, proper utilization of unreliable samples typically leads to more promising results.

**Ablation on representation learning.** Here we ablate the contributions of two components in representation enhancement: mixup augmentation training and consistency regularization. Specifically, we compare SoLar with three variants: 1) *SoLar w/o MU* which removes Mixup augmentation training; 2) *SoLar w/o CR* which removes consistency regularization; 3) *SoLar w/o MU+CR* which removes both Mixup and consistency parts; 4) *PRODEN w/o MU+CR* is the PRODEN algorithm that removes Mixup and consistency parts as well. From Table 10, we can observe that both SoLar and PRODEN benefit from Mixup and consistency regularization techniques. In contrast to the relatively complicated contrastive learning modules in PiCO, which may be not directly suitable for the long-tailed setup, our simpler design can also alleviate the representation dilemma of PLL.

### C.5 Results with Non-Uniform Data Generation

In reality, some labels may be more analogous to the ground-truth than others, and CIFAR100-H-LT is exactly one of the cases. Following [6, 10], we further test SoLar with a non-uniform data generation

Table 10: Ablation results on representation learning techniques.

| Ablation | CIFAR10-LT | | | | CIFAR100-LT | | | |
|---|---|---|---|---|---|---|---|---|
| | All | Many | Med. | Few | All | Many | Med. | Few |
| SoLar | 74.16 | 96.50 | 76.01 | 49.34 | 53.03 | 74.33 | 54.09 | 30.62 |
| CC w/o MU+CR | 36.98 | 79.38 | 30.63 | 3.04 | 25.96 | 47.08 | 23.82 | 7.04 |
| PRODEN w/o MU+CR | 46.61 | 85.43 | 44.65 | 10.40 | 31.78 | 55.09 | 32.38 | 7.85 |
| SoLar w/o MU | 69.40 | 92.77 | 72.75 | 41.58 | 47.41 | 71.45 | 48.06 | 22.70 |
| SoLar w/o CR | 57.97 | 92.78 | 61.05 | 19.05 | 47.74 | 70.18 | 51.85 | 21.06 |
| SoLar w/o MU+CR | 44.83 | 82.33 | 40.39 | 13.25 | 30.88 | 50.52 | 30.53 | 11.61 |

Table 11: Characteristics of the real-world partial label datasets. #Avg. Cand. indicates the average number of candidate labels per sample. Note that some labels in the Lost dataset have no sample at all and hence, we calculate the imbalanced ratio by using the non-zero minimum label count.

| Datasets | #Examples | #Features | #Labels | #Avg. Cand.[†] | Imb. Ratio |
|---|---|---|---|---|---|
| BirdSong | 4,998 | 38 | 13 | 2.18 | 11.33 |
| Lost | 1,122 | 108 | 16 | 2.23 | 40.00 |
| Soccer Player | 17,472 | 279 | 171 | 2.09 | 954.33 |
| Yahoo! News | 22,991 | 163 | 219 | 1.91 | 308.79 |

process, with the following flipping matrix:

$$
\begin{bmatrix}
1 & 0.5 & 0.4 & 0.3 & 0.2 & 0.1 & 0 & \cdots & 0 \\
0 & 1 & 0.5 & 0.4 & 0.3 & 0.2 & 0.1 & \cdots & 0 \\
\vdots & & & & \cdots & & & & \vdots \\
0.5 & 0.4 & 0.3 & 0.2 & 0.1 & 0 & 0 & \cdots & 1
\end{bmatrix},
\tag{16}
$$

where each entry denotes the probability of a label being a candidate. From Figure 3 (c), we evaluate SoLar on CIFAR10-LT with an imbalance ratio $\gamma = 100$. It can be observed that SoLar still outperforms the baselines, which further validates its strong disambiguation ability.

### C.6 Results on Real-world Partial-Label Learning Datasets

In this section, we test the performance of SoLar on four classical real-world datasets[2], including Lost, Bird Song, Soccer Player and Yahoo!News. As shown in Table 11, these datasets are naturally imbalanced, which highlights the motivation of our work. In particular, the Soccer Player dataset has an extremely severe imbalanced ratio of 954.33. Thus, the evaluation protocol of previous works, *i.e.*, uniformly splitting a testing set, is unrealistic on these real-world datasets. To this end, we propose a (roughly) balanced testing set sampling rule as follows: i) if one label is associated with $\geq 200$ data points, we uniformly select 100 samples for testing; ii) otherwise, we uniformly sample half of the data for testing. For performance comparisons, we additionally compare SoLar with two more PLL methods that are tailored for tabular data: IPAL [11] is a graph-based PLL method that propagates candidate labels to recover label confidences; PLDA [12] is a feature-selection-based method that maximizes the mutual-information-based dependency between features and labels and we choose PL-SVM [13] as the base learner. We disable mixup training and consistency regularization as they are not applicable to tabular data. As shown in Table 12, SoLar achieves comparable or better performance to all the baselines under the conventional uniform testing set splitting setup. When the testing set is roughly balanced, SoLar obtains the best performance on all four datasets. These results further highlight the superiority of SoLar on the imbalanced PLL problem.

## D  Societal Impact

In this section, we briefly discuss several societal impacts of our work. The most obvious merit of our study is to reduce the cost of annotation by enabling coarse-grained labeling. This is a

---

[2]http://palm.seu.edu.cn/zhangml/Resources.htm#data

Table 12: Performance comparisons on real-world partial-label learning datasets.

| Methods | Lost | BirdSong | Soccer Player | Yahoo!News |
|---|---|---|---|---|
| | Uniform Testing Set | | | |
| SoLar | $77.86_{\pm 6.36}$ | $\mathbf{72.05_{\pm 1.76}}$ | $\mathbf{57.94_{\pm 1.13}}$ | $67.62_{\pm 0.64}$ |
| VALEN | $74.11_{\pm 4.14}$ | $71.59_{\pm 2.08}$ | $57.16_{\pm 0.62}$ | $67.93_{\pm 0.30}$ |
| PRODEN | $\mathbf{77.98_{\pm 5.83}}$ | $71.81_{\pm 1.52}$ | $57.12_{\pm 1.28}$ | $67.87_{\pm 0.87}$ |
| CC | $77.85_{\pm 5.83}$ | $71.86_{\pm 1.94}$ | $56.44_{\pm 0.90}$ | $\mathbf{67.96_{\pm 0.90}}$ |
| IPAL | $72.50_{\pm 2.92}$ | $70.28_{\pm 1.33}$ | $54.79_{\pm 1.37}$ | $66.50_{\pm 1.05}$ |
| PLDA | $66.07_{\pm 1.89}$ | $67.68_{\pm 1.94}$ | $50.26_{\pm 0.46}$ | $53.66_{\pm 0.99}$ |
| | (Roughly) Balanced Testing Set | | | |
| SoLar | $\mathbf{70.56_{\pm 3.25}}$ | $\mathbf{68.72_{\pm 1.17}}$ | $\mathbf{24.97_{\pm 0.68}}$ | $\mathbf{58.18_{\pm 0.68}}$ |
| VALEN | $60.95_{\pm 2.71}$ | $67.49_{\pm 1.22}$ | $20.56_{\pm 0.79}$ | $56.30_{\pm 0.70}$ |
| PRODEN | $68.85_{\pm 4.17}$ | $67.72_{\pm 1.42}$ | $24.22_{\pm 0.88}$ | $55.98_{\pm 0.67}$ |
| CC | $68.18_{\pm 3.97}$ | $67.78_{\pm 1.35}$ | $23.84_{\pm 1.01}$ | $56.01_{\pm 0.72}$ |
| IPAL | $64.35_{\pm 2.50}$ | $66.82_{\pm 1.49}$ | $11.34_{\pm 0.23}$ | $54.90_{\pm 0.53}$ |
| PLDA | $54.28_{\pm 4.77}$ | $53.24_{\pm 2.08}$ | $16.17_{\pm 0.90}$ | $25.50_{\pm 0.50}$ |

two-edged sword for the community. On the one hand, non-expert annotators can be employed for crowdsourcing labeling. On the other hand, if the partial-label learning paradigm is widely applied, the need for precisely annotated data would be significantly reduced, which may cause potential employment destruction as a consequence of reducing the need for human annotators. Another potential application of our method is data privacy. For instance, we may ask respondents to answer some private information when collecting some survey data. The candidate set-style labeling enables the respondents to exclude several wrong answers, which would be more privacy-friendly.