# OpenReview forum: "SoLar: Sinkhorn Label Refinery for Imbalanced Partial-Label Learning"
_NeurIPS.cc/2022/Conference — NeurIPS 2022 Accept_

### Official Review · Reviewer_nPgC · 2022-07-11

**Rating:** 2
**Confidence:** 5
**Soundness:** 2 fair
**Presentation:** 3 good
**Contribution:** 1 poor

**Summary:**

This paper concentrates on how to perform disambiguation in partial label learning with the long-tailed label distribution considered. The authors propose an Optimal Transport-based framework called SoLar that allows to refine the disambiguated labels towards matching the marginal class prior distribution. They conduct experiments on several benchmarks to demonstrate the superiority of the proposed method.

**Questions:**

The authors should compare more SOTA PLL methods except for some specific pseudo-labeling-based method to verify their point that the performance will degenerate in the long-tailed setting. The authors should also conduct experiments to show that their motivation works on benchmark PLL datasets, such as lost/BirdSong/Soccer Player/Yahoo!News.



**Limitations:**

The motivation is not convincing as stated in the weakness.
The novelty and the techquie seem to be limited.
The experiments have some limitations. For example, experiments on many PLL benchmark datasets (such as lost/BirdSong/Soccer Player/Yahoo!News ) are not conducted, and many SOTA methods are not compared, for example, Instance-Dependent Partial Label Learning, NeurIPS 2021.



**Strengths And Weaknesses:**

Strengths:
The paper is well-written and easy to follow. The proposed method seems to be technically reasonable.

Weaknesses:
The motivation is not convincing. On the one hand, the paper claims that "existing PLL methods have been commonly driven by the assumption that training data consists of class-balanced distribution", which is not true as many SOTA PLL methods do not have the assumption, and they can deal with the imbalanced-class problem natually embedded in the disambiguation. On the other hand, the paper claims that "we find that current best-performing PLL methods display degenerated performance in the long-tailed setting. This happens because the predictions of pseudo-labeling—a core component that PLL methods rely on—can be largely biased towards the head and majority classes." However, many SOTA methods are not pseudo-labeling based methods, and the authors only present one PLL mehtod, i. e., PRODEN, to display the degenerated performance in the long-tailed setting, which is not convincing.

---

> ### Author Response · Authors · 2022-08-02
> **Response to Reviewer nPgC (1/2)**
>
> Thanks very much for your comments and suggestions. Below we address the feedback and comments in detail:
>
> **Q1. The paper claims that "existing PLL methods have been commonly driven by the assumption that training data consists of class-balanced distribution", which is not true as many SOTA PLL methods do not have the assumption, and they can deal with the imbalanced-class problem naturally embedded in the disambiguation.**
>
> **A:** The reviewer raises concern about the above claim. On that, we first want to point out that the most advanced approaches (i.e. the SOTA methods) — including PRODEN, PICO, and newly proposed VALEN [1] — have treated all labels as equal during the core label disambiguation process, which we may deem this as an implicit assumption of class-balancing.
>
> Secondly, while a few prior works may not bear a balanced-class assumption from a methodological perspective, they certainly have not put much emphasis on the class-imbalanced scenarios towards the development of their methods.
>
> Thirdly, ubiquitous with the prior work, this lack of consideration of the class-imbalanced distribution has indeed led to poor performances as shown in our main table in Section 4.
>
> To the best of our knowledge, there is no prior work that has systematically studied PLL under class-imbalanced settings like ours. In this regard, we believe our work can trigger a new round of thinking in the community.
>
> **Q2.The paper claims that "we find that current best-performing PLL methods display degenerated performance in the long-tailed setting. This happens because the predictions of pseudo-labeling—a core component that PLL methods rely on—can be largely biased towards the head and majority classes." However, many SOTA methods are not pseudo-labeling-based methods, and the authors only present one PLL method, i. e., PRODEN, to display the degenerated performance in the long-tailed setting, which is not convincing.**
>
> **A:** Indeed, we demonstrate the degenerated performances of PRODEN qualitatively in Figure 1 as a visual example. However, we kindly disagree with the reviewer because we quantitatively evaluate the degenerated performances of an extensive number of PLL methods, including PiCO (ICLR’22), LWS (ICML’21), PRODEN (ICML’20), and CC (NeurIPS’20), in Table 1, 2, and Figure 1. With these results, we further notably conduct a few-shot experiment, displayed in Table 2, where we evidently locate the poor performance onto the tail labels in the long-tailed setup. Besides, as per the reviewer’s request, we supplement our paper with some newer PLL methods — VALEN [1] (NeurIPS’21) and PLDA [3] (SIGKDD’ 22) — where SoLar still performs at best.
>
> **Q3. The authors should also conduct experiments to show that their motivation works on benchmark PLL datasets, such as lost/BirdSong/Soccer Player/Yahoo!News. The authors should compare more SOTA PLL methods.**
>
> **A:** This is a very helpful suggestion. We chose CIFAR-10, CIFAR-100, and CUB-200 as our benchmarking datasets in our original submission, mainly due to that we closely follow the prior published work in ICLR, NeurIPS, and ICML, such as PiCO paper, PRODEN paper, etc.
>
> However, we do believe that the suggested benchmarks provided by the reviewer are helpful in particular these datasets have a wider spectrum of modalities. We supplement a series of experiments to show the effectiveness of our SoLar approach on these datasets, demonstrated in Appendix C.6. Besides, we find these real-world datasets are all naturally imbalanced, which accords with our motivation that ambiguous labels are more likely to occur on long-tailed datasets (also mentioned in our Conclusion). With almost no change to the SoLar setting, our approach still gets established as the one of most advantageous methods on these supplemented datasets.
>
> Last but not least, as we also mentioned in the previous response, we added the comparison of SoLar with more SOTA PLL methods, including VALEN [1] (NeurIPS’21), IPAL [2], and PLDA [3] (SIGKDD’22). The results can be found in Tables 1, 2, 12, and Appendix C.6, where SoLar maintains its significant advantages.
>
> **References:**
>
> [1] Xu N, Qiao C, Geng X, et al. Instance-dependent partial label learning. Advances in Neural Information Processing Systems, 2021, 34: 27119-27130.
>
> [2] Zhang M L, Yu F. Solving the partial label learning problem: An instance-based approach. Twenty-fourth international joint conference on artificial intelligence, 2015.
>
> [3] Wei Wang and Min-Ling Zhang. Partial label learning with discrimination augmentation. SIGKDD, 2022

---

> > ### Author Response · Authors · 2022-08-02
> > **Response to Reviewer nPgC (2/2) -- The motivation of SoLar**
> >
> > **“Limitation: The motivation is not convincing as stated in the weakness.”**
> >
> > **A:** With the answers we posted above to address the reviewer’s concern, we want to reiterate the motivation to develop the SoLar approach. First, we claim that the real-world PLL scenarios often bear a long-tailed prediction distribution, which is also partially verified by the datasets recommended by the reviewer. Based on this observation, we identify the challenges and the degenerated effect of current PLL methods. In that regard, we intend to establish a novel and challenging scenario, blending PLL and long-tailed distribution, which is very much untapped by the prior work. Therefore, we propose SoLar, an optimal transport-based label refinery method, as a viable solution that demonstrates state-of-the-art performances on a variety of benchmarks.

---

> ### Comment · Area_Chair_xm97 · 2022-08-09
> **Response to author rebuttal?**
>
> Dear reviewer,
>
> It seems that the author rebuttal directly addresses what you have identified as weaknesses of the paper: that the SOTA PLL methods already handle the long-tailed setting adequately, that they only compare to PRODEN empirically. They have also included an appendix showing experimental results on the benchmark PLL datasets you suggested.
>
> Could you please respond to the author's rebuttal, saying whether or not (and why) they have adequately addressed your concerns, and whether your rating of the paper has changed?
>
> Thanks!

---

### Official Review · Reviewer_PJgg · 2022-07-11

**Rating:** 8
**Confidence:** 5
**Soundness:** 4 excellent
**Presentation:** 4 excellent
**Contribution:** 4 excellent

**Summary:**

This work approaches the class-imbalanced partial-label learning problem where each training example is assigned a set of candidate labels. Efforts are made to address two technical challenges. First, even the best-performing PLL method fails to identify the true labels from the candidate labels, because the pseudo-labels are largely biased toward head labels. To address this problem, an optimal transport-based objective is proposed to enforce constraints on pseudo-labels to match the class prior. Second, this work proposes a new class prior estimation algorithm along with a sample selection mechanism for practical training. Experiments validate the effectiveness of the proposed method.

**Questions:**

I noticed that the class prior vector r is initialized as uniform. But, why can't we count the number of candidate labels as a rough estimation of the class prior? It is typically close to the true class prior and may perform better.

**Limitations:**

It would be appreciated if experimental results on large-scale LTL datasets are provided.

Potential negative societal impacts are discussed in the Appendix and the PLL framework may cause potential employment destruction.


**Strengths And Weaknesses:**

Pros:

(1) Most studies about PLL assume the actual label distribution is balanced, which may not hold in practice. This work reveals the leading cause of performance degradation of the SOTA PLL methods long-tailed PLL problem and proposes a principled OT-based framework to resolve this. It is a serious attempt that makes PLL more suitable for real-world applications.

(2) The iterative class prior estimation technique is novel, as estimating the true class prior without an additional validation set is intractable for any weakly-supervised learning problems. Other imbalanced weakly-supervised learning algorithms can potentially benefit from this technique.

(3) Theoretical results show the consistency of the proposed OT objective.

(4) Experiments are thorough and the proposed SoLar algorithm shows impressive performance.



Cons:

(1) The main weakness is the evaluation was conducted on CIFAR datasets. With long-tailed sampling, the resulting data examples are typically sparse. Can SoLar get similar performance improvement on real-world long-tailed datasets?

(2) Typos: 'Goup of Labels' in Figures.

(3) Wrong figure caption in Figure 3, CIFAR-10-LT rather than CIFAR100-H-LT.

---

> ### Author Response · Authors · 2022-08-02
> **Response to Reviewer PJgg**
>
> Thanks very much for your insightful comments and suggestions! We have properly revised our paper based on your reviews and below are our detailed responses.
>
> **Q1. Can SoLar get similar performance improvement on real-world long-tailed datasets?**
>
> **A:** That’s a good point, thank you! We additionally conducted experiments on the SUN397 dataset and we refer the reviewer to Table 4 and Appendix C.1 for more details. The training set contains 88,904 images with an imbalanced ratio of 2311/50=46.22. From Table 4, we can observe that SoLar still outperforms the baselines by a substantial margin. Besides, we also conducted a new series of experiments on classical PLL datasets which are naturally imbalanced; see Appendix C.6. We can observe that SoLar achieves better results than the baselines. These results clearly verify the effectiveness of SoLar on real-world long-tailed datasets.
>
> **Q2. Typos: 'Goup of Labels' in Figures and Wrong figure caption in Figure 3, CIFAR-10-LT rather than CIFAR100-H-LT.**
>
> **A:** We are sorry for the lack of carefulness. We have double-checked our manuscript and fixed all typos and mistakes.
>
> **Q3. Can we count the number of candidate labels as a rough estimation of the class prior?**
>
> **A:**  The answer is affirmative. To verify it clearly, we experimented with SoLar by initializing $r_j$ with the proportion of candidate counts, and the results are reported in Table 3 (the variant *SoLar w Cand-Est*) in the revised version. It can be shown that the performances of *SoLar w Cand-Est* favorably match the original SoLar.

---

> > ### Comment · Reviewer_PJgg · 2022-08-10
> > **After rebuttal**
> >
> > Thank the authors for addressing all my concerns. I think this is a good paper and I will keep my score.

---

### Official Review · Reviewer_HWkc · 2022-07-11

**Rating:** 8
**Confidence:** 5
**Soundness:** 4 excellent
**Presentation:** 4 excellent
**Contribution:** 4 excellent

**Summary:**

This paper proposes an optimal transport framework Solar for imbalanced partial label learning where each data point has multiple candidate labels and the latent label distribution is skewed. The key of Solar is to refine the pseudo-labels by a constraint of within-candidate allocation and class prior matching. In parallel, a moving-average-based class prior algorithm is proposed to generate class prior for the aforementioned constraints. Experiment results on CIFAR-LT and CUB-200 datasets validate the effectiveness of Solar in improving few-shot accuracy.

**Questions:**

Please see my main review.

**Limitations:**

Societal Impacts: The main negative impact is lower annotation costs may decrease the requirement for annotator employment.

Limitations: The experiments need to be further improved.


**Strengths And Weaknesses:**

Strengths:

1.The long-tailed learning problem is important but is generally overlooked in the PLL problem. This work systematically investigated the imbalanced PLL problem by identifying the failure of existing PLL methods and a new Solar framework, which can inspire future research in this field.

2.Solid theoretical analysis is given. The overall objective guarantees to recover the true classifier at a population level. The detailed derivation of the Sinkhorn-Knopp algorithm is also provided.

3.The experiments are sufficient. The ablation study strongly supports the effectiveness of the label refinery procedure. Reproducibility is ensured as the source code was submitted.

Weaknesses:


1.In experiments, the PRODEN method also uses mixup and consistency training techniques for fair comparisons. What about other competitive baselines? I'd like to see how much the strong CC method could benefit from the representation training technique.

2.It is not clear why the proposed sample selection mechanism helps preserve the label distribution.

3.In App. B.2, a relaxed solution of Sinkhorn-Knopp algorithm is proposed. Why the relaxed problem guarantees to converge?Does Solar always run this relaxed version of Sinkhorn-Knopp?

4.How is gamma in the Sinknhorn-Knopp affect the performance?

5.How does the class distribution estimate for PRODEN in Figure 1?

---

> ### Author Response · Authors · 2022-08-02
> **Response to Reviewer HWkc**
>
> We are glad that you enjoyed our work! We have updated our manuscript based on the comments of the reviewers.
>
> **Q1. I'd like to see how much the strong CC method could benefit from the representation training technique.**
>
> **A:** We have tested the performance of all the baselines and a new baseline VALEN [1] with mixup (MU) and consistency training (CT) techniques. The results are reported in Table 1,2 in the revised manuscript (without changing the name of these methods). From the two labels, we can observe that most methods can benefit from the representation learning techniques. But, SoLar still achieves substantially better performance than them, which further highlights the effectiveness of SoLar.
>
> **Q2. It is not clear why the proposed sample selection mechanism helps preserve the label distribution.**
>
> **A:** We apologize for the confusion! Recall that we anchor the total sample number to be $k=\min(\lceil r_j\rho|B|\rceil, |B_j|)$. For the first term, let us consider an ideal setup where both the estimated class prior and model predictions are accurate. Obviously, each label would be assigned $r_j|B|$ examples. Thus, we anchor the first term to be $\rho$-proportional to this ideal setting. The second term indicates that the selected are conducted within the subset $B_j$ where examples have the $j$-th class as their pseudo label assignment. Through the Sinkhorn label refinery procedure, the pseudo labels are enforced to match the class prior $r_i$ and the size of subsets $|B_j|$ roughly follows distribution $r_i$. Thus, we say the proposed sample selection procedure approximately preserves the class prior distribution.
>
> **Q3. Why does the relaxed problem guarantee converge? Does Solar always run this relaxed version of Sinkhorn-Knopp?**
>
> **A:** Note that the modified matrix $M+\epsilon$ is entry-wise positive. Henceforth, the convergence property of our relaxed version of Sinkhorn-Knopp is guaranteed by the Sinkhorn’s Theorem [2] as follows
>
> > If $\mathbf{A}$ is an $n × n$ matrix with strictly positive elements, then there exist diagonal matrices $\mathbf{D}_1$ and $\mathbf{D}_2$ with strictly positive diagonal elements such that $\mathbf{D}_1\mathbf{A}\mathbf{D}_2$ is doubly stochastic. The matrices $\mathbf{D}_1$ and $\mathbf{D}_2$ are unique modulo multiplying the first matrix by a positive number and dividing the second one by the same number.
>
> The Sinkhorn-Knopp algorithm is one classic algorithm to approach this double stochastic matrix with a convergence guarantee [3].
> We believe the unrelaxed version can result in a better pseudo-label assignment. Hence, in our implementation, we run this relaxed version only when the original version fails. Empirically, this relaxed version will be called only in the first few epochs.
>
> **Q4.How is gamma in the Sinknhorn-Knopp affect the performance?**
>
> **A:** If we get the reviewer right, the ''gamma'' here may refer to the ''lambda'' In the Sinkhorn-Knopp algorithm. We add a new set of experiments for ablation on $\lambda$ as shown in Appendix B.5. In general, a too small $\lambda$ results in poor label assignment as the optimal transport objective becomes hard to be resolved. With a too large $\lambda$, our objective is over-smoothed and thus the resultant solution may deviate from the true one, slightly degrading the performance. We empirically found that $\lambda=3$ is a proper choice.
>
> For the $\gamma$ parameter, which controls the skewness of the datasets, we refer the reviewer to Table 1 for details. All methods perform worse when $\gamma$ becomes larger, regarding more severe label skewness. But, our SoLar consistently outperforms under varying $\gamma$ values.
>
> **Q5.How does the class distribution estimate for PRODEN in Figure 1?**
>
> **A:** For PRODEN, we count training samples grouped by the predicted labels (categorical) and report the proportion of the counts. We have updated the caption of Figure 1 to make it more transparent.
>
> **References:**
>
> [1] Xu N, Qiao C, Geng X, et al. Instance-dependent partial label learning[J]. Advances in Neural Information Processing Systems, 2021, 34: 27119-27130.
>
> [2] https://en.wikipedia.org/wiki/Sinkhorn%27s_theorem
>
> [3] Sinkhorn, Richard, & Knopp, Paul. (1967). "Concerning nonnegative matrices and doubly stochastic matrices". Pacific J. Math. 21, 343–348.

---

> > ### Comment · Reviewer_HWkc · 2022-08-08
> > **Thank for the authors' response**
> >
> > My questions are well addressed and also vote for acceptance.

---

### Author Response · Authors · 2022-08-02
**Summary of author response - thank you for the valuable and constructive feedback**

We sincerely appreciate all reviewers for their great efforts in providing constructive and valuable feedback. We are pleased that the reviewers find our work **well-written** (R3), **theoretically solid** (**R1,R2**), and **experimentally comprehensive** (**R1,R2**). Moreover, we are more than encouraged that the reviewers agree that our work makes **an important attempt towards the imbalanced PLL problem** (**R1,R2**).

We have addressed the reviewers’ comments and concerns in individual responses to each reviewer. The reviews allowed us to strengthen our manuscript and the changes made are summarized below:

- [R1] Updated the results of all baselines with mixup and consistency training (See Table 1,2)

- [R1] Added new ablation studies on the $lambda$ parameter (See Appendix C.4 and Figure 3 (b))

- [R1] Revised the caption of Figure 1 for a clear description of distribution estimation

- [R2] Added new results on the real-world long-tailed SUN397 dataset (See Section 4.4 and Appendix C.1)

- [R2] Corrected all the typos and mistakes

- [R2] Added new results on SoLar with candidate counts initialization

- [R3] Added new results on the PLL benchmark datasets (See Appendix C.6)

- [R3] Added two SOTA PLL methods VALEN (NeurIPS’21) and PLDA (SIGKDD’22) for performance comparisons (See Table 1,2 and Appendix C.6)

(* As abbreviations, we refer to reviewers **HWkc** as R1, **PJgg** as R2, and **nPgC** as R3 respectively.)

---

### Author Response · Authors · 2022-08-08
**Please let us know if you have any further questions regarding our rebuttal**

Dear reviewers,

As we move closer to the end of the discussion panel, we haven't heard back from the reviewer. Please let us know if you have any further questions regarding our rebuttal.

Sincerely,

Authors

---

### Meta-Review · Area_Chair_xm97 · 2022-09-01

**Recommendation:** Accept
**Confidence:** Less certain

**Metareview:**

Most approaches to partial-label learning (PLL) tasks assume the label distribution is balanced, which may not hold in practice. This paper provides a principled optimal transport-based framework to resolve the issues with performance degradation caused by skewed label distributions in PLL. The reviewers found that the work makes a theoretically solid and experimentally comprehensive attempt towards solving this practical and under-addressed problem.

**Award:**

No

---

### Decision · Program_Chairs · 2022-09-14

Accept